# *In vivo* assessment of a delayed release formulation of larazotide acetate indicated for celiac disease using a porcine model

Hiroko Enomoto[1], James Yeatts[1], Liliana Carbajal[1], B. Radha Krishnan[2], Jay P. Madan[2], Sandeep Laumas[2], Anthony T. Blikslager[1], Kristen M. Messenger[1]*

1 Comparative Medicine Institute, North Carolina State University, Raleigh, NC, United States of America,
2 Innovate Biopharmaceuticals Inc., Raleigh, NC, United States of America

* kmmessen@ncsu.edu

**Data Availability Statement:** All relevant data are within the manuscript and its Supporting Information files.

## Abstract

There is no FDA approved therapy for the treatment of celiac disease (CeD), aside from avoidance of dietary gluten. Larazotide acetate (LA) is a first in class oral peptide developed as a tight junction regulator, which is a lead candidate for management of CeD. A delayed release formulation was tested *in vitro* and predicted release in the mid duodenum and jejunum, the target site of CeD. The aim of this study was to follow the concentration versus time profile of orally administered LA in the small intestine using a porcine model. A sensitive liquid chromatography/tandem mass spectrometry method was developed to quantify LA concentrations in porcine intestinal fluid samples. Oral dosing of LA (1 mg total) in overnight fasted pigs resulted in time dependent appearance of LA in the distal duodenum and proximal jejunum. Peak LA concentrations (0.32–1.76 µM) occurred at 1 hour in the duodenum and in proximal jejunum following oral dosing, with the continued presence of LA (0.02–0.47 µM) in the distal duodenum and in proximal jejunum (0.00–0.43 µM) from 2 to 4 hours following oral dosing. The data shows that LA is available in detectable concentrations at the site of CeD.

## Introduction

Celiac disease (CeD) is one of the most common autoimmune disorders affecting around 1% of the population worldwide [1, 2]. There has been a notable rise in the prevalence of CeD in the last 50 years and a rise in the rate of diagnosis in the last 10 years [2]. CeD is defined as a chronic small intestinal immune mediated enteropathy that is precipitated by exposure to dietary gluten, which is broken down into immunologically active gliadin fragments in genetically predisposed individuals. Gliadin indirectly stimulates the secretion of zonulin from the lamina propria of the intestine into the intestinal lumen, which leads to the binding of zonulin to purported apical receptors of the enterocyte. This initiates a complex series of tight junction events that involves phosphorylation of tight junction proteins, which induces a loss of epithelial barrier function, exacerbating innate and adaptive immune responses [1, 3, 4]. Untreated and partially treated CeD is associated with an increased risk for multiple comorbidities, such as diarrhea, abdominal pain, infertility, osteoporosis, joint pain, arthritis, uveitis, cataracts,

**Funding:** The study was funded by Innovate Biopharmaceuticals (Jay Madan), which is now 9meters (https://9meters.com/). KMM and AB received the award (TSA #91690). Innovate Biopharmaceuticals (now 9meters) provided support in the form of salary for authors BRK, TPM, and SL. The funders had no role in study design, data collection and analysis, decision to publish, or preparation of the manuscript. The specific roles of these authors are articulated in the 'author contributions' section.

**Competing interests:** The authors have read the journal's policy and have the following competing interests: AB and KM's institution has received funding from Innovate Biopharmaceuticals, now 9meters Biopharm. KM has received speaking honoraria, travel compensation, or research support from Zoetis, Bayer, Ellevet, Piedmont Animal Health, Scullion Strategy, Jurox, Mallinckrodt, and RxActuator. AB has consulted for Innovate Biopharma and 9-meters Biopharma. He has also received honoraria from Kemin. RK, JM and SL are currently employees or consultants of or own stock in 9Meters Biopharm.This does not alter our adherence to PLOS ONE policies on sharing data and materials. There are no patents, products in development or marketed products associated with this research to declare.

alopecia areata, neuropathies, and lymphomas [2, 4–7]. Currently, a gluten free diet (GFD) is the only available approach to manage symptoms in CeD patients, but it is not completely effective due to hidden gluten from food contamination in kitchens, restaurants, and during food processing [4]. Recurrent CeD signs and symptoms resulting from inadvertent or deliberate gluten exposure have been reported in approximately 70% of CeD patients on a GFD [8, 9]. Additionally, refractory celiac disease is unresponsive to the treatment even with a strict gluten free diet and no other effective treatment has been established [7, 10, 11]. There is therefore an unmet need for non-dietary therapies for the management of CeD.

Larazotide acetate (LA) is an orally administered, locally acting, synthetic eight amino acid peptide that is known to act as a tight junction regulator [12]. LA acts as a zonulin inhibitor, capable of closing leaky or open interepithelial junctions, thereby having the potential to prevent exposure to gliadin [13]. Presently, LA is being studied in phase 3 clinical trials using a delayed release formulation designed to reach the target site for treatment of CeD in the proximal small intestine [12, 14–16].

No study has been conducted to show the presence of LA and/or its fragments (Fig 1A and 1B) in the gastrointestinal tract (GIT) upon dosing. While earlier studies attempted (unsuccessfully) to determine LA pharmacokinetics from plasma, the peptide is broken down in the small intestine and there is no systemic absorption of LA or the fragments, therefore collection of intestinal fluid directly from the intestinal tracts is ultimately necessary to determine the presence, concentrations, and duration of LA at its site of action [17]. However, several anatomic and physiologic characteristics of the intestinal tract make frequent sampling challenging. Repeated aspiration of the gastrointestinal fluid increases the risk of peritonitis. Furthermore, it is impossible to repeatedly aspirate from the same anatomic location due to GI motility, and abdominal aspiration is invasive and painful to the subject. The sacrifice of multiple animals for collection of intestinal fluid contents does not allow for repeated sampling in the same animal. *In vivo* ultrafiltration (UF) is a minimally invasive method to collect and determine protein unbound drug concentrations in animal models [18, 19]. The probe fibers are made of a semipermeable dialysis membrane allowing water, electrolytes and low weight molecules (<30,000 Daltons) to pass into the collection system. A vacutainer attached to the assembly passively aspirates fluid over time and allows for the collection of multiple samples without causing distress to the animal. UF probes were previously successfully implanted into the intestinal tract of calves, which allowed for safe and effective continuous sampling of intestinal fluid [20]. UF probes have been used to collect several drug compounds from various tissue sites in animals [21–25], but to our knowledge, this technique has never been applied to the collection of LA and its fragments directly from the intestinal tract. The purpose of this study was to (1) evaluate the dissolution profile of LA in the simulated media and to quantify

**Fig 1.** Amino acid sequence of larazotide acetate (a) and its fragments (b) Larazotide acetate: H-Gly-Gly-Val-Leu-Val-Gln-Pro-Gly-OH (a) Fragment 1: H-Gly-Val-Leu-Val-Gln-Pro-Gly-OH; Fragment 2: H-Val-Leu-Val-Gln-Pro-Gly-OH; Fragment 3: H-Gly-Gly-Val-Leu-Val-Gln-Pro-OH; Fragment 4: H-Val-Leu-Val-Gln-Pro-OH.

the concentrations of LA in the porcine small intestine, and to (2) evaluate the use of UF as a method for intestinal fluid collection in a porcine model. We hypothesized that UF could be used to evaluate concentrations of LA directly at the intestinal site of action.

## Materials and methods

### *In vitro* dissolution experiment

1 mg LA capsules were dissolved in the simulated gastric media and simulated intestinal media, and percent dissolved LA were calculated at appropriate time points. The dissolution method was performed according to the current USP dissolution monograph chapter 711 [26]. The analytical method was validated following the FDA Bioanalytical Method Validation Guidelines for Industry [27]. The detailed methods are shown in S1 Text.

**LA bead composition.**   In this study, a delayed release formulation of LA was tested. This is the formulation of LA that is being tested in the phase 3 clinical trials. The capsule contained a mixture of two types of beads (A and B) with two different thicknesses of enteric coating to trigger release in the mid duodenum and complete release within the proximal jejunum. The enteric coating included an enteric methacrylate polymer, eudragit, which is well known as an enteric coating polymer that delays the release of the active substance in the acidic surroundings of the stomach. The enteric coating was designed to release LA at pH level above pH 5, which would include release in the duodenum and jejunum over the course of about 3 hours.

### *In vivo* ultrafiltration experiment

**Animals and housing.**   This study was carried out in accordance with the recommendations in the Guide for the Care and Use of the Laboratory Animals [28] and approved by the Institutional Animal Care and Use Committee at North Carolina State University (protocol number: 18-154-B). Three female Yorkshire cross pigs weighing between 15–20 kg and 6–8 weeks of age were used in this study. Pigs were obtained from North Carolina State University's commercial herd. All pigs were considered healthy on the basis of physical exams, which were performed upon delivery and after a minimum of 3 days' acclimatization. The animals were housed individually in stainless steel pens to allow comprehensive observation of changes to the animal's behavior, feed consumption, and volume or character of excreta. While housed in pens, each animal had free access to water, and commercial pig pelleted feed was provided twice daily. Housing was controlled at temperature 17.8–28.9°C, humidity 30–70% and an alternating 12 hour light/dark cycle was maintained and air flow was at least 10 air changes per hour with 100% fresh air. On the last day of the study, each pig was sedated with a xylazine/ketamine combination, then humanely euthanized by administration of an overdose of sodium pentobarbital intravenously. Once death was confirmed by lack of a heart beat and corneal reflex, a post mortem exam was performed to confirm the location of UF probes in the intestines during the duration of the study period.

**Surgical procedure for ultrafiltration probe placement.**   Food and water were withheld from pigs at least 12 hours prior to anesthesia and surgery. Pigs were sedated with injection of ketamine (11 mg/kg, IM; Vedco) and xylazine (0.25 mg/kg, IM; Akorn Animal Health). Intravenous administration of buprenorphine (0.04 mg/kg; Par Pharmaceutical) and intramuscular administration of flunixin meglumine (2.2 mg/kg; Norbrook) and ceftiofur sodium (Excede for Swine, 8mg/kg; Med Pharmex Inc) were given for analgesia and infection prophylaxis, respectively. Pigs were intubated and maintained under general anesthesia using isoflurane delivered in 100% oxygen during the surgery. The abdomen was approached via midline laparotomy, and the pylorus was located. The duodenum was traced from the pylorus to an accessible point on the left side of the abdomen approximately 15–18 cm distal to the pylorus (Fig

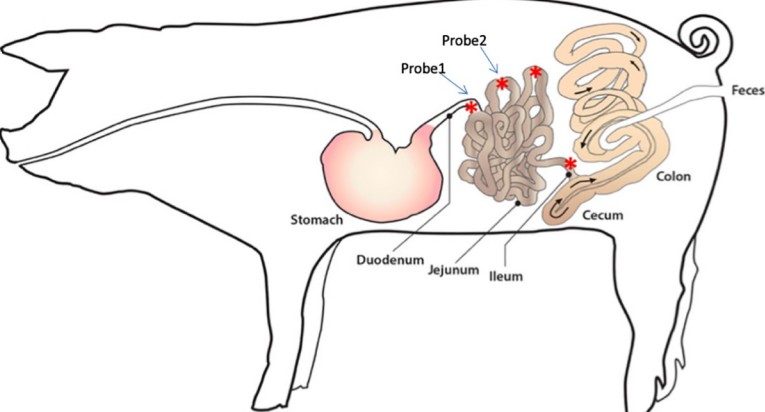

**Fig 2. The location of implantation of ultrafiltration probes.** The location of the first probe placement was 15–18 cm distal to the pylorus, the distal duodenum. The location of the second probe placement was 20 cm distal to the duodenum, the proximal jejunum.

2). The intestine was lifted and an incision was made using a #11 scalpel blade, piercing the anti-mesenteric intestinal wall into the lumen. A guide needle was used to introduce the UF probe into the duodenum threading the probe proximally toward the pylorus. The guide needle was subsequently removed, taking care to leave the UF probe in place [20]. The three loops of the UF probe remained within the intestine, and each probe was secured in the intestine by using a Halstead pattern suture around the probe, followed by a finger trap suture pattern using 3–0 Vicryl (Henry Schein). The nonpermeable connection tubing of the UF probe was extended to the lateral body wall, paramedian to the incision, and retrieved through a guide needle in order to exteriorize the tubing. The tubing was secured outside the animal by suturing the tubing to the skin using 3–0 nylon suture (Fig 3). The tubing was attached to a needle inserted into a vacutainer for sample collection during the remainder of the experiment. A second UF probe was placed into the jejunum 20 cm distal to the duodenum using the same methodology (Fig 2). The abdomen was closed in a routine manner. The porcine gastrointestinal model shown in Fig 3. The pigs were monitored for assessment of pain each day of the experiment and given buprenorphine as an analgesic medication if necessary.

**Drug formulation, administration and intestinal fluid collection.** The delayed release formulation of LA was provided by Innovate Biopharmaceuticals Inc. (currently, 9Meters Biopharm Inc., Raleigh, NC, USA). This formulation was identical to the formulation tested in the *in vitro* experiment as previously described. The formulated beads were weighed and placed into gelatin capsules prior to dosing. Pigs were fasted overnight for approximately 14 hours; water was withheld for 2 hours prior to dosing. On the morning of each experiment, a single capsule (1 mg) of LA, approximate dose of 0.05 mg/kg, or a placebo capsule was administered to each animal by mouth, then chased with 120 mL water. The animals were observed to ensure swallowing of the entire capsule. After drug or placebo administration, water was withheld for a further 2 hours, and food for four hours. Intestinal fluid (S1 Fig) was collected into a 3mL vacutainer (Becton Dickinson) via the UF probes every 1 hour over a 4 hour period on each day of the study (Fig 4). Each vacutainer was prefilled with 100 μL of a quenching solution (5% trifluoroacetic acid (TFA) in 80% acetonitrile (ACN):15% water) to prevent degradation of LA by intestinal enzymes during the collection period. Following sampling, an additional amount of quenching solution was added to the sample if necessary based on the sample volume collected to maintain an approximately consistent ratio of intestinal fluid: quench solution (250: 40). Samples were centrifuged at 13,200 *x g* for 5 minutes at 4˚C. The resulting

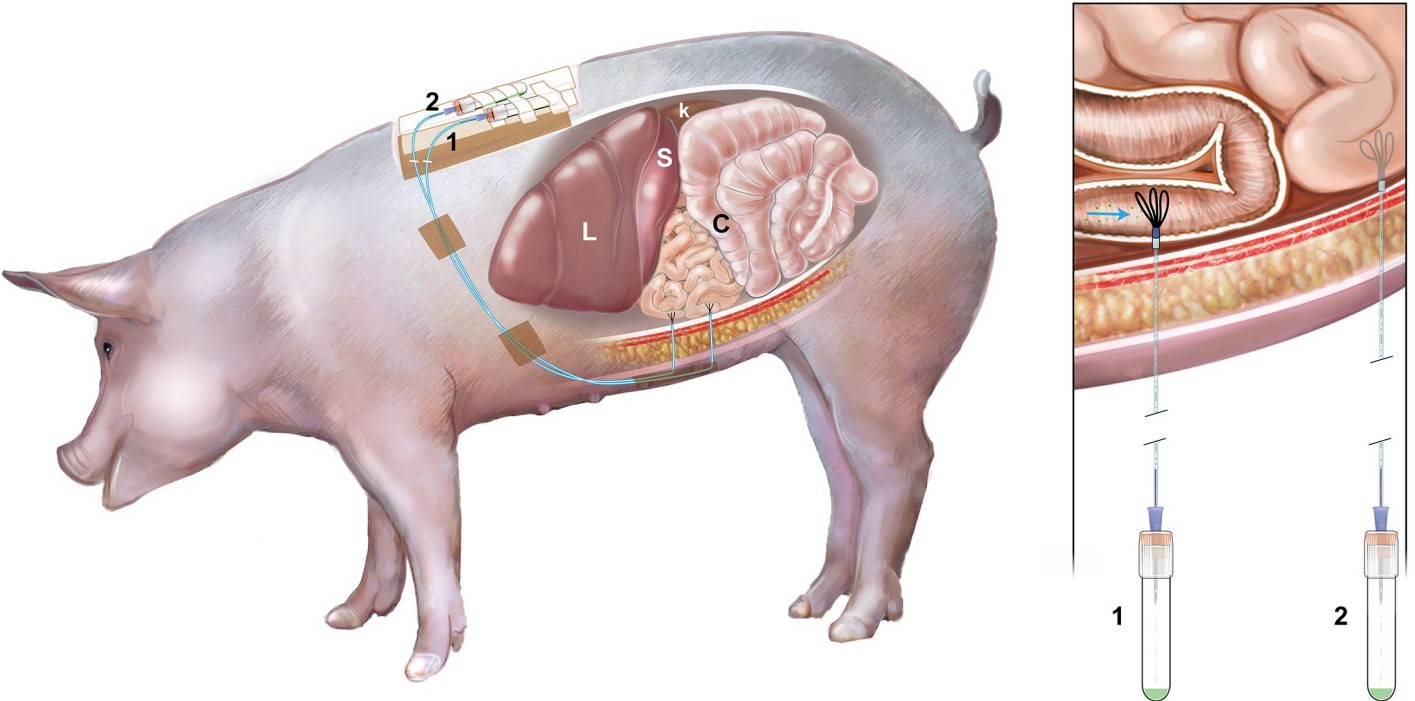

**Fig 3. The porcine gastrointestinal model.** This model was implanted the ultrafiltration probes (UF) at duodenum and jejunum and the intestinal fluid was collected via UF probes continuously. L: liver, S: spleen, K: kidney, C: colon, 1: first UF probe, 2: second UF probe.

supernatant was removed and stored at -80˚C until analysis by ultra performance liquid chromatography and tandem mass spectrometry (UPLC-MS/MS).

**Chemicals and reagents.** All reagents were of LC/MS grade. ACN, formic acid and TFA were purchased from Fisher Chemical (Raleigh, NC, USA). LA and its fragments (1, 2, 3 and 4) standards were provided by Innovate Biopharmaceuticals Inc. (Currently, 9Meters Biopharm Inc).

**Instrumentation.** Analysis of LA and fragments were carried out via ultra performance liquid chromatography (UPLC) and tandem mass spectrometric (MS/MS) detection (Waters Corporation, Milford, MA). The UPLC-MS/MS system consisted of a Waters Acquity UPLC I class Binary Solvent Manager, Acquity UPLC Sample Manager FTN and a Xevo TQD tandem mass spectrometer (Waters Corporation, Milford, MA).

**Preparation of standard solutions.** The 3.5, 7, 14, 35, 70, 140 and 350 μM standards of LA were prepared by dilution of mixed stock solutions of LA (700 μM) and its fragments 1–4 (750, 800, 750 and 850 μM respectively) with diluent of 1:1 ACN: water. The standards of fragment 1–4 were made from mixed stock solution at the range of 3.75–375, 4.0–400, 3.75–375 and 4.25–425 μM respectively.

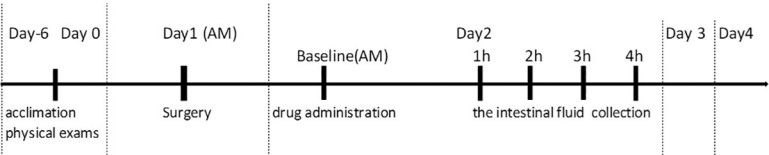

**Fig 4. A timeline of the study design.** Overnight fasted pigs received 1 mg larazotide acetate (or placebo) and the intestinal fluid was collected every hour for four hours via the ultrafiltration probe on each experimental day.

**Calibration standard preparation.** Each standard solution (3.5, 7, 14, 35, 70, 140 and 350 μM) of LA was diluted with the blank intestinal fluid to give concentrations 0.035, 0.07, 0.14, 0.35, 0.7, 1.4 and 3.5 μM for the calibration curve. The range of the calibration curve for fragment 1–4 was 0.038–3.75, 0.04–4.0, 0.038–3.75, and 0.04–4.25 μM respectively. The blank intestinal fluid and dilution solvent were injected with every batch.

**Sample preparation.** The samples were centrifuged at 10,000 g for 5 minutes. The supernatant was filtered through a 0.2 μm polyvinylidene difluoride membrane filter directly into Waters maximum recovery sample vial (Waters Corporation, Milford, MA) prior to injection into the UPLC-MS/MS system. Quality control standards were injected with every batch.

**UPLC- MS/MS conditions.** Chromatographic separation was performed by a gradient elution on the ACQUITY UPLC BEH C18 1.7 μm column (2.1 mm x 50 mm) with VanGuard Pre-column (Waters Corporation, Milford, MA). The mobile phase solvents were 0.02% TFA in water (A) and 0.02% TFA in ACN (B) at a flow rate of 0.4 mL/min for 8 minutes. The gradient program mobile phase conditions were 95% of A and 5% of B for the first 5.5 minutes, then changed linearly to 10% of A and 90% of B from 5.5–6.5 minutes, then immediately back to 95% of A and 5% of B from 6.51–8 minutes to re-equilibrate at the initial conditions. The column temperature was 35˚C, the autosampler temperature was maintained at 4˚C and the injection volume was 7 μL. The positive electrospray ionization mode (ESI (+)) was used with the multiple reactions monitoring (MRM). The tune page source voltages were 2.0 kV and 47 V for the capillary and cone respectively. The source desolvation temperature was 600˚C. The source desolvation gas flow was 750 L/hr and the cone gas was 50 L/hr. The source temperature was 150˚C. The MS file cone voltage (V) settings for LA, fragments 1, 2, 3, and 4 were 44, 42, 42, 40, and 38 V with collision energy settings (V) of 60, 58, 54, 62, and 54 V respectively. Argon was used as the collision gas and nitrogen as the desolvation and cone gases. Quantification was performed using the transitions Parent (m/z) 726.65, 669.54, 612.58, 669.54 and 555.64 for LA, fragment 1, 2, 3 and 4 respectively and Daughter (m/z); 72.08, 86.14, 72.1, 72.1 and 72.1 for LA, fragment 1, 2, 3 and 4 respectively with the retention time 2.33, 2.32, 2.23, 2.43 and 2.35 minutes for LA, fragment 1, 2, 3 and 4 respectively (S1 Table). The chromatogram of LA is shown in S2 Fig.

**The lower limit of quantification and the lower limit of detection.** Lower limit of quantification (LLOQ) was defined as the lowest concentration that produced a peak area 5 times the blank solvent peak area, had an accuracy within 20% of the nominal value, and a precision of no more than 20% CV. The LLOQ of the LA was 0.07 μM (0.05 μg/mL). The LLOQ of the fragment 1, 2, 3 and 4 was 0.15, 0.16, 0.15 and 0.17 μM (0.1 μg/mL) respectively. The lower limit of detection (LLOD) was the lowest concentration that produced a peak area > 3 times blank solvent peak area. The LLOD of the LA was 0.0175 μM (0.0125 μg/mL). The LLOD of the fragment 1, 2, 3 and 4 was 0.0024, 0.0050, 0.0188 and 0.0213 μM (0.0016, 0.0031, 0.0012 and 0.0125 μg/mL) respectively.

**Calibration curve.** The calibration curves of LA and its fragment 1–4 were fit with a weighted (1/concentration) linear equation. The calibration range of 0.035–3.5 μM (LA), 0.038–3.75 μM (fragment 1 and 3), 0.04–4.0 μM (fragment 2), 0.04–4.25 μM (fragment 4) was linear with a coefficient of determination, $R^2$, greater than or equal to 0.99. Each calibration standard concentration could be back calculated to within 15% of the true concentration (S3 Fig).

**Precision and accuracy.** A total 6 replicate samples at low, medium and high concentrations (0.21, 1.05 and 2.8 μM of LA, 0.45, 1.13 and 3 μM of fragment 1, 0.48, 1.2 and 3.2 μM of fragment 2, 0.45, 1.13 and 3 μM of fragment 3 and 0.51, 1.28 and 3.4 μM of fragment 4) were tested on 3 days and interday and intraday precision and accuracy were calculated. Mean of each concentration were within 15% of the nominal value and have a precision not exceeding 15% CV (S2 Table).

**Pharmacokinetics.** Non compartmental pharmacokinetic analyses of LA in the intestinal fluid was performed using commercially available software (Phoenix® WinNonlin® Software version 8.3, Certara, Princeton, NJ). The pharmacokinetic parameters estimated for LA in intestinal fluid after oral administration included the elimination rate constant ($\lambda z$), elimination half life ($HL_{\lambda z}$), the area under the curve from time zero to the last time point ($AUC_{last}$), the maximum concentration (Cmax), time to maximum concentration (Tmax), the mean drug residence time from time zero to the last time point ($MRT_{last}$), which were calculated using the linear log trapezoidal method.

## Results

### *In vitro* experiment

Data was collected from 6 capsules. Average (± SD) dissolution (%) are shown in Table 1 and Fig 5. The *in vitro* dissolution data predicted that the delayed release formulation of LA was not dissolved in the simulated gastric dissolution media for 2 hours. Dissolution was 39.7–42.8%, 79.1% and 93.7% at 30–60 minutes, 90 minutes and 120–180 minutes in intestinal dissolution media.

### *In vivo* experiment

A pilot study was initially performed with one pig to determine the feasibility of intestinal fluid collection. Sample data collected from this pilot study was not pooled with the study data. Thus, two animals were included in the study with one dosing event per day per animal. Each animal was used for up to four dosing events (either LA or placebo) without any signs of adverse effects. In the two experimental animals, the UF probes continuously collected intestinal fluid over the entire study period. Average (± SD) volume of fluid collected each hour at the distal duodenum was 604 ±259, 835 ±72, 515 ±75 and 487 ±113 μL at 1, 2, 3, and 4 hour time points, respectively. Average (± SD) volume of fluid collected each hour at the proximal jejunum was 720 ±70, 575 ± 80, 765 ±190 and 556 ± 250 μL at the 1, 2, 3, and 4 hour time points respectively (Table 2 and S3 Table).

Oral dosing of the clinical formulation of LA (1mg) in overnight fasted pigs resulted in time dependent appearance of LA in the distal duodenum and the proximal jejunum. Peak LA

**Table 1. *In vitro* dissolution data of 1 mg larazotide acetate of a delayed release formulation in 6 capsules.**

|  | Sampling time points (minutes) | Time in GIT (minutes) | Average (±SD) Dissolution (%) of 1 mg LA (n = 6) |
|---|---|---|---|
| Simulated gastric media (Stomach) | 0 | 0 | 0 |
|  | 30 | 30 | 0 |
|  | 60 | 60 | 0.1 ± 300 |
|  | 120 | 120 | 0.7 ± 85.7 |
| Simulated intestinal media (Small intestine) | 15 | 135 | 1.4 ± 42.9 |
|  | 30 | 150 | 39.7 ± 11.4 |
|  | 45 | 165 | 41.2 ± 14.4 |
|  | 60 | 180 | 42.8 ± 12.6 |
|  | 90 | 210 | 79.1 ± 18.3 |
|  | 120 | 240 | 93.7 ± 2.4 |
|  | 180 | 300 | 93.7 ± 2.5 |

Times (minutes) in small intestine were back calculated by adding 120 minutes to sampling time point respectively. LA: larazotide acetate, GIT: gastrointestinal tract, SD: standard deviation

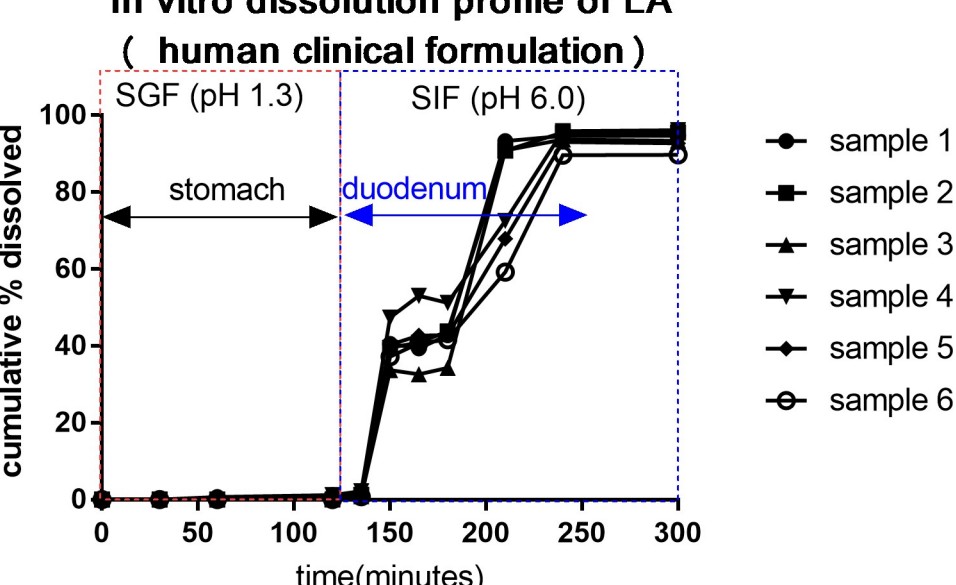

**Fig 5. The *in vitro* dissolution profile of larazotide acetate (LA) of the delayed release formulation.** The *in vitro* dissolution data predicts the delayed release formulation of LA will begin releasing in the mid duodenum and complete release in the proximal jejunum. SGF: simulated gastric fluid, SIF: simulated intestinal fluid. Samples 1–6 were replicated and included 1 mg of LA respectively.

concentrations ranged from 0.32–1.76 μM, and were noted at 1 hour in the distal duodenum and the proximal jejunum. The continued presence of LA was detected in distal duodenum for 2–4 hours (0.02–0.47 μM), and in proximal jejunum for 2–4 hours (0.00–0.43 μM). The LA concentrations were below LOD in one sample at the 3 hour time point and in 2 samples at the 4 hour time point in the proximal jejunum. The LA concentrations of these samples were reported as zero. The average (± standard deviation) concentrations of LA at 1, 2, 3 and 4 hours are shown in Fig 6, Table 2 and S4 Table.

The LA was not detected in the placebo (control) group throughout the study (Table 2 and S4 Table).

The presence of LA fragments (1, 2, 3 and 4) were detected in duodenal and jejunal samples. However, all concentrations were below the LLOQ of 0.15, 0.16, 0.15 and 0.17 μM respectively.

The results of the pharmacokinetic analysis are presented in Table 3.

## Discussion and conclusions

This study was designed to assess the *in vivo* delivery and gastrointestinal transit profile of the delayed release formulation of LA intended for use in CeD patients using a porcine model. The results of the animal experiments show that concentrations of LA were present in the distal duodenum for the entire four hours and proximal jejunum for three hours (4 hours in one sample) following oral administration of the delayed release formulation *in vivo*. Since CeD patients have mucosal morphological changes localized to the upper small intestine [6, 7], this delayed release formulation is ideal to target diseased tissue in the duodenum and proximal jejunum. This is the first study to quantify the presence of LA in the small intestine *in vivo* over a recommended dosing interval, although numerous previous studies have confirmed pharmacodynamic effects [2, 4, 29–31].

The GIT comprises rather complex biochemical and physiological process such as enzyme, luminal pH, body temperature, peristalsis movement, gastric and intestinal residence time and

**Table 2. Concentration data of larazotide acetate in the intestinal samples upon oral administration of 1mg larazotide acetate in a delayed release formulation in pigs.**

| Time (hour) | Probe Location | Average (±SD) Concentration of LA (μM) in intestinal fluid (n = 3) | Average (±SD) Volume (mL) of intestinal fluid (n = 3) |
|---|---|---|---|
| Clinical formulation | Distal duodenum | | |
| 0 | | 0.00 ± 0.00 | 2583 ± 93 |
| 1 | | 0.74 ± 0.30 | 604 ± 259 |
| 2 | | 0.27 ± 0.14 | 835 ± 72 |
| 3 | | 0.12 ± 0.09 | 515 ± 75 |
| 4 | | 0.03 ± 0.01 | 487 ± 113 |
| | Proximal jejunum | | |
| 0 | | 0.00 ± 0.00 | 2364 ± 621 |
| 1 | | 1.10 ± 0.48 | 720 ± 70 |
| 2 | | 0.25 ± 0.13 | 575 ± 80 |
| 3 | | 0.09 ± 0.10 | 765 ± 190 |
| 4 | | 0.03 ± 0.04 | 556 ± 250 |
| Placebo control | Distal duodenum | | |
| 0 | | 0.00 ± 0.00 | 1848 ± 929 |
| 1 | | 0.00 ± 0.00 | 827 ± 113 |
| | Proximal jejunum | | |
| 0 | | 0.00 ± 0.00 | 1751 ± 373 |
| 1 | | 0.00 ± 0.00 | 943 ± 148 |

1μM of larazotide acetate (LA) = 0.71 μg/ml, SD: standard deviation

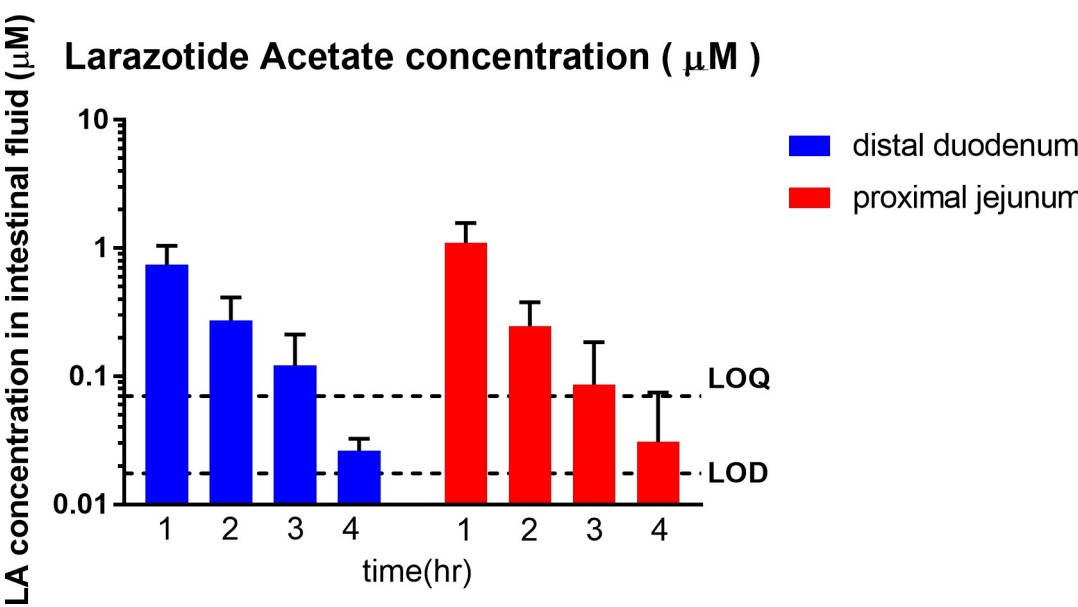

**Fig 6. Concentration v time profile of larazotide acetate (LA, μM) in the porcine intestinal fluid analyzed via UPLC-MS/MS (n = 3).** Time dependent appearance of LA in the distal duodenum and the proximal jejunum was observed. Peak LA concentrations ranged from 0.32 to 1.76 μM and were first observed at the 1 hour time point in the distal duodenum and the proximal jejunum. The highest concentration of LA was detected in the proximal jejunum at 1 hour. The continued presence of LA was detected in the distal duodenum (0.02–0.47 μM) and proximal jejunum (0.00–0.43 μM) from 2 to 4 hours post administration.

**Table 3. The pharmacokinetic parameters for larazotide acetate in the intestinal fluid after oral larazotide acetate administration (1 mg total; approximate dose 0.05 mg / kg).**

| Distal duodenum | Dose Replicate | | | | | |
|---|---|---|---|---|---|---|
| Parameter (unit) | # 1 | # 2 | # 3 | Mean | SD | CV % |
| $\Lambda z$ (1/h) | 0.79 | 1.23 | 1.41 | 1.14 | 0.32 | 28.1 |
| $HL_{\lambda z}$ (h) | 0.88 | 0.56 | 0.49 | 0.65 | 0.21 | 32.2 |
| Tmax (h) | 1 | 1 | 1 | 1 | 0 | 0 |
| Cmax (µg / mL) | 0.23 | 0.70 | 0.67 | 0.53 | 0.26 | 49.3 |
| $AUC_{last}$ (µg·h /mL) | 0.36 | 0.83 | 1.13 | 0.77 | 0.39 | 50.2 |
| $MRT_{last}$ (h) | 1.58 | 1.42 | 1.65 | 1.55 | 0.12 | 7.80 |
| Proximal jejunum | | | | | | |
| $\Lambda z$ (1/h) | 1.39 | 0.90 | 0.69 | 1.0 | 0.36 | 36.2 |
| $HL_{\lambda z}$ (h) | 0.50 | 0.77 | 1 | 0.76 | 0.25 | 33.3 |
| Tmax (h) | 1 | 1 | 1 | 1 | 0 | 0 |
| Cmax (µg / mL) | 0.46 | 1.26 | 0.63 | 0.78 | 0.42 | 53.4 |
| $AUC_{last}$ (µg·h /mL) | 0.52 | 1.22 | 0.99 | 0.91 | 0.36 | 39.4 |
| $MRT_{last}$ (h) | 1.31 | 1.26 | 1.53 | 1.37 | 0.14 | 10.3 |

$\lambda z$: elimination rate constant, $HL_{\lambda z}$: elimination half life, Tmax: time to the maximum concentration, Cmax: maximum concentration, $AUC_{last}$: area under the curve from time zero to the last time point, $MRT_{last}$: mean residence time from time zero to the last time point

1 µg / mL = 1.4 µM. SD: standard deviation, CV: coefficient of variation

luminal composition; thus it is more complex than the *in vitro* static dissolution model [32]. An *in vivo* model is more ideal to study the true movement and release of a drug formulation. In this study, a porcine model was established for *in vivo* human oral drug assessments for several reasons. In particular, pigs have similar intestinal anatomy and physiology to that of human beings [33]. The porcine small intestinal pH has a similar range to that of humans [33–36]. More specifically, the fasted pig's gastric pH is 1.2–4.0 and the proximal small intestinal pH is 6.7. The fasted human gastric pH is 1.0–3.5 and 6.0–7.0 for the proximal small intestine [33–36]. The variation in gastrointestinal volume, osmolality and intestinal transit time are other important determinants of drug release and absorption [36, 37]. The transit time from mouth through the small intestine is reported to be similar in pigs and humans, taking 3–4 hours in pigs as compared to 2–4 hours in humans in the fasted state [33].

Following oral administration, the parameters having the most influence on the drug dissolution are the physical and chemical characteristics of the dosing formulation [38–41]. The enteric methacrylate polymer, eudragit, is derived from esters of acrylic and methacrylic acid by free radical polymerization. The use of eudragit for targeted drug release and eudragit's stability in gastric fluid in the presence of digestive enzymes are well known [39, 42]. This *in vitro* study confirmed the stability of LA in the simulated gastric fluid (dissolved LA % was 0.7%) and dissolution of LA in simulated media in time dependent manner. LA solubility in the digestive fluid depends primarily on time and luminal pH [37]. The variability of drug transit time, luminal pH and other factors such as gastrointestinal fluid volume may have contributed to the variability of the LA concentrations in the small intestine.

UF probes were successfully placed in two different locations in the small intestine in all three pigs. Although unanticipated complications occurred in the initial pilot pig (loss of the UF probe from within the intestinal lumen), the UF probes were well tolerated at both duodenal and proximal jejunal sites in the remaining two pigs. Using the probes, intestinal fluid was collected over four hours, continuously from the exact same location, without invasive

sampling or euthanasia, allowing for significant reduction in the number of experimental animals, and minimization of intra-animal variation in the data. The UF probes exclude all proteins and molecules larger than 30,000 Daltons and allows the collection of a "clean" ultrafiltrate that can be analyzed for drug concentrations or even biomarkers of drug efficacy. Warren et al., reported that intestinal fluid could be collected from the ileum and spiral colon of steers over 48 hours with an 88% success rate. Occasional absence of sample in the collection tube was noted as a possible complication, attributable to clogging of the pores on the fragile membrane of the UF or to damage of the membrane due to the motility of the intestinal tract and ingesta [20]. Although some samples were not obtained in the pilot pig, all samples at all time points were collected in the other two animals. In another study in horse hooves, 33% of the UF probe sites became infected, which was attributed to inadequate preparation of the probe placement site, insufficient maintenance of sterility during placement, or prolonged probe placement time [23]. In the present study, no signs of infection were observed; the probes were placed aseptically and tissues remained healthy over the 5 day study duration. Overall, it appears that UF probes can be used at multiple sites in the intestinal tract of pigs to collect gastrointestinal fluids. Continued refinement of optimal UF probe placement and maintenance techniques are necessary for future studies on gastrointestinal drug pharmacokinetics.

*In vitro* dissolution is the standard method used in the pharmaceutical industry to assess drug release from solid oral dosage forms and to predict the release profile in the GI tract by simulating the GI environment with appropriate buffer media. *In vivo/ in vitro* correlation analysis (IVIVC) was specifically designed to assess whether *in vitro* predictions match *in vivo* drug behavior. However, in this study IVIVC could not be performed for several reasons, including the lack of an injectable formulation, an immediate release and other multiple other extended release formulations [43] and the inability to quantify the total volume of the intestinal fluid as the UF probes only collected a small fraction of the entire volume. However, we described the pharmacokinetics of LA at two different sites in the small intestine in the pigs in this study. The pharmacokinetics of LA have not been described in people, despite the compound undergoing multiple clinical trials to date. Paterson et al., reported that oral LA (12mg) could be detected in the human plasma by HPLC-MS/MS, but all plasma concentrations were below LLOQ (0.5 ng/ml) [17]. Leffler et al., reported that the concentrations of LA and metabolites in human plasma were below the LOQ (0.5 ng/ml) in all groups administered 0.25, 1, 4 and 8 mg LA [16], thus LA pharmacokinetics could not be determined in human plasma. Since the present study only included the small number of pigs, data interpretation was limited. Additionally, our UPLC-MS/MS method was not sensitive enough to quantify the larazotide fragments that were identified. Lastly, healthy pigs were used in this study. Drug release and pharmacokinetics could be different in patients with CeD or other gastrointestinal pathology [44]. In spite of these limitations, the data showed that the LA was detectable in the small intestine after dosing, and an animal model was described to study drugs that act locally in the intestinal tract.

Despite the limitations discussed, this was the first study in which the UF probes were successfully placed in the intestinal tract of pigs to obtain concentrations of LA directly from the site of action. Our technique may be useful in future pharmacokinetic studies to analyze other formulations of LA or other locally acting drugs in the small intestine.

## Supporting information

**S1 Text. The in vitro dissolution test methods.**
(PDF)

**S1 Fig. Intestinal fluid collection system.**
(TIF)

**S2 Fig. A typical chromatogram of larazotide acetate at 1.38 μM.**
(TIF)

**S3 Fig. Calibration curves of larazotide acetate and its fragment 1–4.**
(TIF)

**S1 Table. Detection of larazotide acetate and its fragments in porcine intestinal fluid matrix by UPLC-MS/MS.**
(PDF)

**S2 Table. Analytical method suitability study data.**
(PDF)

**S3 Table. Individual intestinal fluid volume (μL) collected via ultrafiltration probes.**
(PDF)

**S4 Table. Individual concentration of larazotide acetate (μM) in the intestinal samples for the delayed release formulation and placebo administration.**
(PDF)

## Acknowledgments

We are grateful to Alice MacGregor Harvey for the medical illustrations.

## Author Contributions

**Conceptualization:** B. Radha Krishnan, Anthony T. Blikslager, Kristen M. Messenger.

**Data curation:** Hiroko Enomoto, James Yeatts, Liliana Carbajal, B. Radha Krishnan, Anthony T. Blikslager, Kristen M. Messenger.

**Formal analysis:** Hiroko Enomoto, James Yeatts, B. Radha Krishnan, Kristen M. Messenger.

**Funding acquisition:** B. Radha Krishnan, Jay P. Madan, Sandeep Laumas, Anthony T. Blikslager.

**Investigation:** Hiroko Enomoto, Liliana Carbajal, Anthony T. Blikslager, Kristen M. Messenger.

**Methodology:** Hiroko Enomoto, James Yeatts, Liliana Carbajal, B. Radha Krishnan, Kristen M. Messenger.

**Project administration:** Hiroko Enomoto, Liliana Carbajal, Anthony T. Blikslager, Kristen M. Messenger.

**Resources:** B. Radha Krishnan, Jay P. Madan, Sandeep Laumas.

**Supervision:** B. Radha Krishnan, Jay P. Madan, Anthony T. Blikslager, Kristen M. Messenger.

**Validation:** Hiroko Enomoto, James Yeatts.

**Writing – original draft:** Hiroko Enomoto, James Yeatts, Anthony T. Blikslager, Kristen M. Messenger.

**Writing – review & editing:** Hiroko Enomoto, Liliana Carbajal, B. Radha Krishnan, Jay P. Madan, Sandeep Laumas, Anthony T. Blikslager, Kristen M. Messenger.

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
