## [Decision Letter · Decision Letter 0]

18 Aug 2020

PONE-D-20-20338

In-vivo assessment of a delayed-release clinical formulation of larazotide acetate indicated

for celiac disease using a novel porcine model

PLOS ONE

Dear Dr. Messenger,

Thank you for submitting your manuscript to PLOS ONE. After careful consideration, we feel that it has merit but does not fully meet PLOS ONE’s publication criteria as it currently stands. Therefore, we invite you to submit a revised version of the manuscript that addresses the points raised during the review process.

We look forward to receiving your revised manuscript.

Kind regards,

Vivek Gupta

Academic Editor

PLOS ONE

Journal Requirements:

"The study was funded by Innovate Biopharmaceuticals (Jay Madan), which is now 9meters (https://9meters.com/). KMM and AB received the award (TSA #91690). Yes, the funder played a role in the study: Study design,review of the data and the manuscript. "

We note that one or more of the authors have an affiliation to the commercial funders of this research study : Innovate Biopharmaceuticals.

2.1. Please provide an amended Funding Statement declaring this commercial affiliation, as well as a statement regarding the Role of Funders in your study. If the funding organization did not play a role in the study design, data collection and analysis, decision to publish, or preparation of the manuscript and only provided financial support in the form of authors' salaries and/or research materials, please review your statements relating to the author contributions, and ensure you have specifically and accurately indicated the role(s) that these authors had in your study. You can update author roles in the Author Contributions section of the online submission form.

2.2. Please also provide an updated Competing Interests Statement declaring this commercial affiliation along with any other relevant declarations relating to employment, consultancy, patents, products in development, or marketed products, etc.  

Reviewers' comments:

Reviewer's Responses to Questions

**Comments to the Author**

1. Is the manuscript technically sound, and do the data support the conclusions?

Reviewer #1: No

Reviewer #2: Partly

2. Has the statistical analysis been performed appropriately and rigorously? 

Reviewer #1: No

Reviewer #2: I Don't Know

3. Have the authors made all data underlying the findings in their manuscript fully available?

Reviewer #1: Yes

Reviewer #2: No

4. Is the manuscript presented in an intelligible fashion and written in standard English?

Reviewer #1: No

Reviewer #2: No

5. Review Comments to the Author

Reviewer #1: The manuscript is not very well articulated. Needs extensive English language revision and making sentences meaningful, short, and formal. Lacks scientific correlation and does not follow or state the regulatory guidelines followed throughout the study. The title does not correlate to the study performed and reported. No proper references are cited. The aims and objectives are poorly defined. A major revision is required and cannot be accepted in the current state. The following are the recommendation the authors need to address and make suitable changes.

1. The abstract seems too descriptive; the authors are advised to make the abstract short and concise.

2. The term “novel’ for the porcine model used is ambiguous, the literature review does show other porcine models for this type of study. The authors need to show ceratin validation of this “novel” method developed or used in this study. This is “novel” accounted if the authors have independently published this porcine protocol and got approval from regulatory agencies or cited by fellow researchers after using this reported method. A clear justification is requested. If the authors want to report only the novel porcine model and analytical method developed then the title has to be changed suitably.

3. The authors are advised not to define what methodologies were followed or developed in the abstract making it verbose, instead, those can be reported as results are highly recommended. Mere mentioning about first development or use of any method in abstract does not increase the scientific quality or acceptance.

4. Including abstract the entire manuscript is recommended for extensive English grammar revision, making sentences formal and direct and not either way.

5. From the abstract, the readers would get confused, whether the author's would like to elevate the porcine model or absorbance of delayed LA formulation in s.intestine- contradicts with aim defined in the abstract. Kindly rephrase.

6. The conclusion in the abstract is poorly defined. Kindly rewrite.

7. Keywords need to be rechecked.

8. The sentence LA is being studied in phase 3 clinical trials using a clinical formulation designed to reach the target site for treatment of CeD in the small intestine [1]”- the reference cited seems no connection to this sentence. Kindly recheck and cite appropriately.

9. The introduction is poorly constructed. The objective defined at the end again contradicts with title and abstract written. How can we use a non-validated animal model for IVIVC? As per which guidelines the authors have performed the comparison? Kindly cite or remove the phrase. Complete revision is highly recommended.

10. The surgical procedures lack citing the references from where at least the method was inspired or developed.

11. The drug formulation lacks to disclose the type of polymers or excipients used to delay or make a trigger release in the duodenum. In-vitro dissolution procedure is not reported or cited. The authors need to carefully address this lacuna.

12. Authors have failed to mention or cite using or as per which regulatory guidelines the analytical method was developed and validated. If validated why all parameters were disclosed.

13. Apart from indicating concentration at distal or proximal duodenum by using analytical method, there is no substantial evidence provided by authors on explaining why the formulation could have restricted and had triggered at a specific site. The authors could have mentioned the polymer or excipient used for mucoadhesiveness, formulate into tablet and take x-ray images to locate and see the presence of the beads and then quantify the LA. This seems to be a more justified approach. The authors need to clarify on this.

14. There are various contamination and infections borne while using probes. The authors did not mention how they have taken measures to minimize this. Authors need to address this.

15. The study has only two studies- one analytical method to quantify LA and porcine model to collect sampling. The authors have failed to constructively frame the IVIVC. The authors re advised to follow regulatory guidelines while drawing a comparison study.

Reviewer #2: In this manuscript, the authors report an in vivo procedure for real-time sampling of intestinal fluid from pigs to profile the concentration of larazotide acetate (LA) for the first time. They developed a sophisticated and sensitive UPLC method for LA quantification at biorelevant concentrations. The in vivo procedure and analytical methods are well-described.

While this in vivo sampling technique holds multiple advantages, the authors need to better elaborate on the correlation of their findings with conventionally used in vitro dissolution studies. This study would be a good addition to the existing literature, following few major revisions as listed below:

Introduction:

1. Along with the publications [14-17] already cited by the authors, it may be worthwhile to include another reference utilizing a very similar in vivo study for intestinal tract measurements in pigs for a different orally administered drug:

Yan, L., Xie, S., Chen, D. et al. Pharmacokinetic and pharmacodynamic modeling of cyadox against Clostridium perfringens in swine. Sci Rep 7, 4064 (2017). https://doi.org/10.1038/s41598-017-03970-9

2. While the introduction provides a good overview of the need for a minimally invasive, real-time sampling method for in vivo GI profiling of an oral drug, it does not provide a concrete conclusion about the stated hypothesis. It will be helpful to add 2-3 sentences describing how (qualitatively and/or quantitatively) the in vitro-in vivo data correlated based on the demonstrated results.

Materials and methods:

1. Line 125: butterfly ‘valve (is missing)’

2. Line 153-154: For reproducibility, it will be helpful to mention the value of consistent ratio maintained for the intestinal fluid and quench solution

3. System suitability:

- Need to rephrase this section as the sentences are confusing due to repeated occurrences of the phrase ‘system suitability’.

- For complete definition of method, please include the standard concentration that was selected for these studies

-What quantitative criteria were set to determine the system suitability/ repeatability?

Results:

1. Reorganize Table 1 to include the volume of intestinal fluid collected from duodenum and jejunum at different time points, respectively and the corresponding LA concentrations

2. Table 1/Fig. 9: It is unclear how n=3 is obtained, if only 2 pigs were used for the study. Are these biological replicates or technical replicates?

Discussion and conclusions:

1. The authors need to revise the discussion sections to improve the flow of concepts and connectivity.

For e.g. The statement in line 262 ‘It has been reported…’ fits better in lines 310/311

2. Line 260: The use of adjective ‘meaningful’ is very vague. It is unclear if they want to imply bio-relevance or agreement with qualitative in vivo release profile, etc.

3. While the authors repeatedly state that the in vivo concentrations correlate well with the in vitro dissolution results, there is no explicit description on how these in vitro results were captured, interpreted and (qualitatively) compared/correlated. It will be helpful to provide a detailed discussion on Fig 3 and its implications for Fig 9.

Figures:

Figure 1 and 2 can be clubbed together

Figure 3:

- Is the in vitro dissolution data published in previous literature, or generated for this study? If reusing from previous literature, please cite the source. If the study was performed for this manuscript, include the methodology and result discussion for the same.

- Edit axes titles to be more descriptive of the units shown: e.g. X axis- time (minutes), Y-axis- cumulative % dissolved

- It is unclear how samples 1-6 are related to each other. Are they replicates? Please edit the legend appropriately.

Figure 5: Elaborate this caption to describe the model and provide legend for notations used (L, S, k, 1,2 etc.)

Figure 6: can be clubbed with Fig. 5 or moved to supplementary information

Figure 7: Multiple spelling errors

Figure 8: can be moved to supplementary information

Figure 9:

- Edit axes titles to be more descriptive of the units shown: e.g. X axis- time (hr), Y-axis- LA concentration in intestinal fluid (μM)

- (Optional) Can also include a similar dotted line to identify LOD

Supplementary information:

(Optional) Include calibration curves and method suitability study data

Copyediting recommendations:

- Need to standardize (italicize and/or remove hyphen) the format of words like ‘in vitro’ and ‘in vivo’, ‘et al’ throughout the document.

- Avoid unnecessary capitalization (e.g. line 135: In-vitro, line 461: Pig)

- Maintain a single space between values and their unit of measurements for standardization (e.g. line 29: ‘1 mg’ instead of ‘1mg’) throughout the manuscript

- Line 91: Capitalization of the letter ‘I’ in the word- institutional

- Line 215: Explicitly define ‘CV’ (as coefficient of variation?)

6. PLOS authors have the option to publish the peer review history of their article (what does this mean?). If published, this will include your full peer review and any attached files.

Reviewer #1: No

Reviewer #2: No

---

## [Author Response · Author response to Decision Letter 0]

10 Dec 2020

Point-by-point Rebuttal PONE-D-20-20338

“In-vivo assessment of a delayed release formulation of larazotide acetate indicated for celiac disease using a porcine model”

We thank both reviewers for their time and constructive comments to strengthen our manuscript. This document addresses each Reviewer’s concerns. The revised manuscript contains highlighted text for the significant additions to the manuscript and used “track changes” to show deleted text. We look forward to your response. 

Sincerely,

Kristen Messenger

2.1. Please provide an amended Funding Statement declaring this commercial affiliation, as well as a statement regarding the Role of Funders in your study. If the funding organization did not play a role in the study design, data collection and analysis, decision to publish, or preparation of the manuscript and only provided financial support in the form of authors' salaries and/or research materials, please review your statements relating to the author contributions, and ensure you have specifically and accurately indicated the role(s) that these authors had in your study. You can update author roles in the Author Contributions section of the online submission form.

Answer : We have amended this section and included the information in our Cover Letter. This study was funded by Innovate Biopharmaceuticals Inc., which is now 9meters Biopharma. Funding included salary support for authors (KM and AB). RK, JM, and SL were employees of Innovate Biopharmaceuticals and are now either employees or consultants for 9meters. Innovate Biopharmaceuticals (RK, JM, and SL) was involved in the study design, data collection, data analysis, decision to publish, and preparation of the manuscript.

Answer: See above. 

2.2. Please also provide an updated Competing Interests Statement declaring this commercial affiliation along with any other relevant declarations relating to employment, consultancy, patents, products in development, or marketed products, etc. 

Answer: Updated and included in the Cover Letter

Answer: We have provided all the data in Supplementary Files. 

Reviewer 1.

1. The abstract seems too descriptive; the authors are advised to make the abstract short and concise.

Answer: Thank you very much for your advice. We shortened the abstract. 

2. The term “novel’ for the porcine model used is ambiguous, the literature review does show other porcine models for this type of study. The authors need to show ceratin validation of this “novel” method developed or used in this study. This is “novel” accounted if the authors have independently published this porcine protocol and got approval from regulatory agencies or cited by fellow researchers after using this reported method. A clear justification is requested. If the authors want to report only the novel porcine model and analytical method developed then the title has to be changed suitably.

 Answer: We have removed “novel” from the model description and title. 

3. The authors are advised not to define what methodologies were followed or developed in the abstract making it verbose, instead, those can be reported as results are highly recommended. Mere mentioning about first development or use of any method in abstract does not increase the scientific quality or acceptance.

 Answer: The abstract has been revised as suggested. 

4. Including abstract the entire manuscript is recommended for extensive English grammar revision, making sentences formal and direct and not either way.

 Answer: All has been revised.

5. From the abstract, the readers would get confused, whether the author's would like to elevate the porcine model or absorbance of delayed LA formulation in. intestine- contradicts with aim defined in the abstract. Kindly rephrase.

 Answer: Abstract has been revised. 

6. The conclusion in the abstract is poorly defined. Kindly rewrite.

 Answer: Revised.

7. Keywords need to be rechecked.

 Answer: We checked and revised them. Key words are now celiac disease, larazotide acetate, porcine, ultrafiltration 

8. The sentence LA is being studied in phase 3 clinical trials using a clinical formulation designed to reach the target site for treatment of CeD in the small intestine [1]”- the reference cited seems no connection to this sentence. Kindly recheck and cite appropriately.

 Answer: The citation has been updated.

9. The introduction is poorly constructed. The objective defined at the end again contradicts with title and abstract written. How can we use a non-validated animal model for IVIVC? As per which guidelines the authors have performed the comparison? Kindly cite or remove the phrase. Complete revision is highly recommended.

 Answer: We removed the phrase of in vivo/ in vitro comparison and revised the conclusion, and revised the entire introduction. As you will read in the Discussion, IVIVC could not be performed for this study since only a single formulation was tested and there is no completely dissolved formulation of LA to compare with, although future studies could focus on different formulations and IVIVC could be more useful.

10. The surgical procedures lack citing the references from where at least the method was inspired or developed.

 Answer: We added a citation on a bovine GI model from which the method was inspired. 

11. The drug formulation lacks to disclose the type of polymers or excipients used to delay or make a trigger release in the duodenum. In-vitro dissolution procedure is not reported or cited. The authors need to carefully address this lacuna.

Answer: The manuscript has been updated. For the reviewer, the strategy for the manufacturing of the larazotide acetate delayed release capsules involved four layers applied over inert sugar cores. The first layer applied is a drug layer and the second layer applied is a sub coat layer. The third layer applied is an enteric coated layer and the fourth layer is a topcoat layer.

Enteric coat layer 

This layer includes an enteric methacrylate polymer, eudragit, which is well known as an enteric coating polymer that delays the release of the active substance in the acidic surroundings of the stomach. The enteric coat layer enables to release Larazotide at pH level above pH 5 thus allow release in the duodenum as well as jejunum and ileum over the course of about 3 hours when the product exposed to simulated intestinal fluid. 

The release rate is tested in vitro by an in house dissolution test method simulating the gastric fluid and the intestinal fluid. The development of the dissolution test method was based on the physicochemical in vitro and in vivo characteristics of the product considering the mechanism of release. 

The in vitro dissolution test for Larazotide is capable of:

● discriminating between batches with respect to critical process parameters which may have an impact on the desired bioavailability. 

● testing for batch to batch consistency of pivotal clinical, bioavailability and routine production batches.

● determining stability of the relevant release characteristics of the product over the proposed shelf life and storage conditions. 

The specification parameters set for dissolution testing are:

In simulated gastric fluid:

● After 120 minutes not more than 10 % of the labeled amount of Larazotide should dissolve in gastric fluid. 

In simulated intestinal fluid

● Not less than 85 % of the labeled amount of Larazotide should dissolve in 120 minutes.

12. Authors have failed to mention or cite using or as per which regulatory guidelines the analytical method was developed and validated. If validated why all parameters were disclosed.

Answer: The method was validated by showing intraday and interday precision and accuracy according to the FDA Bioanalytical Method Validation Guidelines for Industry. We uploaded the method suitability study data to the supplementary file. 

13. Apart from indicating concentration at distal or proximal duodenum by using analytical method, there is no substantial evidence provided by authors on explaining why the formulation could have restricted and had triggered at a specific site. The authors could have mentioned the polymer or excipient used for mucoadhesiveness, formulate into tablet and take x-ray images to locate and see the presence of the beads and then quantify the LA. This seems to be a more justified approach. The authors need to clarify on this.

Answer: We have now included the in vitro dissolution test developed to serve as a surrogate marker for in vivo behavior and thereby confirm consistent therapeutic performance of batches from routine production. We don’t think that a pig abdominal radiograph would have sufficient resolution to see the beads, although this is a consideration for future studies. We think that measuring the concentration of larazotide acetate directly from different sites in the intestine is important to see larazotide on the surface of the intestinal membrane at the target site in dynamic environment. 

14. There are various contamination and infections borne while using probes. The authors did not mention how they have taken measures to minimize this. Authors need to address this.

Answer: We performed the routine surgical preparation of the abdomen by using chlorhexidine/alcohol alternating scrubs, preoperative administration of antibiotics (ceftiofur sodium 8.0 mg/kg), used sterile ultrafiltration probes and close observation of the incision daily for signs of infection, including heat, pain, swelling or discharge from the surgical incision. Our lab has extensive experience using these probes and to date we have not had any serious issues with contamination or infections while using the probes as directed. 

15. The study has only two studies- one analytical method to quantify LA and porcine model to collect sampling. The authors have failed to constructively frame the IVIVC. The authors are advised to follow regulatory guidelines while drawing a comparison study.

Answer: We could not perform the IVIVC because this study did not have the immediate release formulation and/or other multiple formulations for testing and validation of the model. This study design was not enough to compare in vivo data to in vitro data. Thus, we removed the phrase “in vivo data corresponded to in vitro data”.

Reviewer 2.

Introduction

1. Along with the publications [14-17] already cited by the authors, it may be worthwhile to include another reference utilizing a very similar in vivo study for intestinal tract measurements in pigs for a different orally administered drug:

Yan, L., Xie, S., Chen, D. et al. Pharmacokinetic and pharmacodynamic modeling of cyadox against Clostridium perfringens in swine. Sci Rep 7, 4064 (2017). https://doi.org/10.1038/s41598-017-03970-9

Answer: Thank you very much for your advice. We added this paper to the reference although it is important to note that this reference refers to ex-vivo model.

2. While the introduction provides a good overview of the need for a minimally invasive, real-time sampling method for in vivo GI profiling of an oral drug, it does not provide a concrete conclusion about the stated hypothesis. It will be helpful to add 2-3 sentences describing how (qualitatively and/or quantitatively) the in vitro-in vivo data correlated based on the demonstrated results.

Answer: Thank you for the advice. We discussed IVIVC with our coauthors, and discovered we could not perform the IVIVC because this study did not have the immediate release formulation and/or other multiple formulations for testing. This study design was not enough to compare in vivo data to in vitro data. We removed the phrase “in vivo data corresponded to in vitro data”.

Material and methods

1. Line 125: butterfly ‘valve (is missing)’

Answer: We revised the description of how we secured the outer tubing to the animal.

2. Line 153-154: For reproducibility, it will be helpful to mention the value of consistent ratio maintained for the intestinal fluid and quench solution

Answer: We added the ratio for the intestinal fluid and quenching solution (250:40).

3. System suitability:

-Need to rephrase this section as the sentences are confusing due to repeated occurrences of the phrase ‘system suitability’. 

- For complete definition of method, please include the standard concentration that was selected for these studies.

- What quantitative criteria were set to determine the system suitability/ repeatability?

Answer: This section has been revised. It now reads: A total 6 replicate samples at low, medium and high concentrations (0.21, 1.05 and 2.8 µM of LA, 0.45, 1.13 and 3 µM of fragment 1, 0.48, 1.2 and 3.2 µM of fragment 2, 0.45, 1.13 and 3 µM of fragment 3 and 0.51, 1.28 and 3.4 µM of fragment 4) were tested on 3 days and interday and intraday precision and accuracy were calculated. Mean of each concentration were within 15 % of the nominal value and have a precision not exceeding 15 % CV (Table S2). 

We uploaded method suitability study data to the supplementary files. 

Results

1. Reorganize Table 1 to include the volume of intestinal fluid collected from duodenum and jejunum at different time points, respectively and the corresponding LA concentrations

Answer: We added the volume of intestinal fluid collected from duodenum and jejunum at each time points respectively. In addition, we added the raw data to the supplementary file.

2. Table 1/Fig. 9: It is unclear how n=3 is obtained, if only 2 pigs were used for the study. Are these biological replicates or technical replicates?

Answer: We dosed one pig twice with 48 hours washout (n=2). The second pig had a single dose (n=1). 

Discussion and conclusions:

1. The authors need to revise the discussion sections to improve the flow of concepts and connectivity.

For e.g. The statement in line 262 ‘It has been reported…’ fits better in lines 310/311

Answer: We revised the discussion.

2. Line 260: The use of adjective ‘meaningful’ is very vague. It is unclear if they want to imply bio-relevance or agreement with qualitative in vivo release profile, etc.

Answer: We removed this adjective.

3. While the authors repeatedly state that the in vivo concentrations correlate well with the in vitro dissolution results, there is no explicit description on how these in vitro results were captured, interpreted and (qualitatively) compared/correlated. It will be helpful to provide a detailed discussion on Fig 3 and its implications for Fig 9.

Answer: See previous explanations on why IVIVC was not performed, which we also included in the revised Discussion.

Figures

Figure 1 and 2 can be clubbed together

Answer: Since figure 1 shows amino acid sequence of LA and fragments and figure 2 shows in vivo ultrafiltration probe, we want to keep these figures separately. 

Figure 3:

Is the in vitro dissolution data published in previous literature, or generated for this study? If reusing from previous literature, please cite the source. If the study was performed for this manuscript, include the methodology and result discussion for the same as.

Answer: We added in vitro methodology, results and discussion.

Edit axes titles to be more descriptive of the units shown: e.g. X axis- time (minutes), Y-axis- cumulative % dissolved

Answer: We changed the description of the title of X axis and Y axis. 

It is unclear how samples 1-6 are related to each other. Are they replicates? Please edit the legend appropriately.

Answer: The six capsules are replicated. 

Figure 5: Elaborate this caption to describe the model and provide legend for notations used (L, S, k, 1, 2 etc.)

Answer: We added the description of this model and legend for all notations.

Figure 6: can be clubbed with Fig. 5 or moved to supplementary information

Answer: We moved figure 6 to the supplementary information. 

Figure 7: Multiple spelling errors

Answer: We corrected the mistakes such as acclimation and intestinal.

Figure 8: can be moved to supplementary information

Answer: We moved figure 8 to the supplementary information. 

Figure 9:

-Edit axes titles to be more descriptive of the units shown: e.g. X axis-time (hr), Y-axis- LA concentration in intestinal fluid (µM) 

- (Optional) Can also include a similar dotted line to identify LOD

Answer: We changed the description of the title of X axis and Y axis and added an LOD line.

Supplementary information:

(Optional) Include calibration curves and method suitability study data

Answer: We added calibration curves (S3 figure) and method suitability study data (S2 table) to supplementary information.

Copyediting recommendations:

- Need to standardize (italicize and/or remove hyphen) the format of words like ‘in vitro’ and ‘in vivo’, ‘et al’ throughout the document.

- Avoid unnecessary capitalization (e.g. line 135: In-vitro, line 461: Pig)

- Maintain a single space between values and their unit of measurements for standardization (e.g. line 29: ‘1 mg’ instead of ‘1mg’) throughout the manuscript

- Line 91: Capitalization of the letter ‘I’ in the word- institutional

- Line 215: Explicitly define ‘CV’ (as coefficient of variation?) 

Answer: All recommendations have been accepted and corrected.

CV stands for coefficient of variation.

---

## [Decision Letter · Decision Letter 1]

13 Jan 2021

PONE-D-20-20338R1

In vivo assessment of a delayed release formulation of larazotide acetate indicated for celiac disease using a porcine model

PLOS ONE

Dear Dr. Messenger,

Thank you for submitting your manuscript to PLOS ONE. After careful consideration, we feel that it has merit but does not fully meet PLOS ONE’s publication criteria as it currently stands. Therefore, we invite you to submit a revised version of the manuscript that addresses the points raised during the review process.

We look forward to receiving your revised manuscript.

Kind regards,

Vivek Gupta

Academic Editor

PLOS ONE

Reviewers' comments:

Reviewer's Responses to Questions

**Comments to the Author**

1. If the authors have adequately addressed your comments raised in a previous round of review and you feel that this manuscript is now acceptable for publication, you may indicate that here to bypass the “Comments to the Author” section, enter your conflict of interest statement in the “Confidential to Editor” section, and submit your "Accept" recommendation.

Reviewer #1: All comments have been addressed

Reviewer #2: All comments have been addressed

2. Is the manuscript technically sound, and do the data support the conclusions?

Reviewer #1: Partly

Reviewer #2: Yes

3. Has the statistical analysis been performed appropriately and rigorously? 

Reviewer #1: Yes

Reviewer #2: I Don't Know

4. Have the authors made all data underlying the findings in their manuscript fully available?

Reviewer #1: Yes

Reviewer #2: Yes

5. Is the manuscript presented in an intelligible fashion and written in standard English?

Reviewer #1: Yes

Reviewer #2: Yes

6. Review Comments to the Author

Reviewer #1: The following recent works should be cited and discussed in the results & discussion, this would elevate the importance of the study.

Expert opinion on drug discovery 14, no. 10 (2019): 957-968.

Pharmaceutics 12, no. 11 (2020): 1124.

Gut and liver 9, no. 1 (2015): 28.

Pharmaceutics 12, no. 7 (2020): 652.

Expert Opinion on Pharmacotherapy 12, no. 11 (2011): 1731-1744.

Processes 8, no. 3 (2020): 316.

Gastroenterology 148, no. 7 (2015): 1311-1319.

Peptides 35, no. 1 (2012): 86-94.

Reviewer #2: (No Response)

7. PLOS authors have the option to publish the peer review history of their article (what does this mean?). If published, this will include your full peer review and any attached files.

Reviewer #1: No

Reviewer #2: No

---

## [Author Response · Author response to Decision Letter 1]

27 Feb 2021

We thank the Reviewer for the additional reference suggestions and have included them in our Introduction and Discussion sections, where relevant. The references have been included in our Bibliography.

---

## [Editor Report · Decision Letter 2]

15 Mar 2021

In vivo assessment of a delayed release formulation of larazotide acetate indicated for celiac disease using a porcine model

PONE-D-20-20338R2

Dear Dr. Messenger,

We’re pleased to inform you that your manuscript has been judged scientifically suitable for publication and will be formally accepted for publication once it meets all outstanding technical requirements.

Kind regards,

Vivek Gupta

Academic Editor

PLOS ONE

---

## [Editor Report · Acceptance letter]

30 Mar 2021

PONE-D-20-20338R2 

In vivo assessment of a delayed release formulation of larazotide acetate indicated for celiac disease using a porcine model

Dear Dr. Messenger:

I'm pleased to inform you that your manuscript has been deemed suitable for publication in PLOS ONE. Congratulations! Your manuscript is now with our production department. 

Kind regards, 

on behalf of

Dr. Vivek Gupta 

Academic Editor

PLOS ONE